# Validation of reference gene stability for miRNA quantification by reverse transcription quantitative PCR in the peripheral blood of patients with COVID-19 critical illness

Amanda Formosa[1,2]*, Erica Acton[2,3], Amy Lee[3], Paul Turgeon[4], Shehla Izhar[2], Pamela Plant[2], Jim N. Tsoporis[2], Sabri Soussi[1,2], Uriel Trahtemberg[2,6], Andrew Baker[1,2,5,7], Claudia C. dos Santos[1,2,5,7,8]*

1 Interdepartmental Division of Critical Care Medicine, Temerty Faculty of Medicine, University of Toronto, Toronto, Canada, 2 The Keenan Research Centre for Biomedical Sciences, Unity Health Toronto, Toronto, Ontario, Canada, 3 Molecular Biology & Biochemistry Department, Simon Fraser University, Burnaby, British Columbia, Canada, 4 Department of Laboratory Medicine and Pathobiology, University of Toronto, Toronto, Ontario, Canada, 5 Department of Critical Care, St. Michael's Hospital, Unity Health Toronto, Toronto, Ontario, Canada, 6 Critical Care Department, Galilee Medical Center, Nahariya, Israel, 7 Institute of Medical Sciences, Temerty Faculty of Medicine, University of Toronto, Toronto, Ontario, Canada, 8 Department of Physiology, Faculty of Medicine, University of Toronto, Toronto, Ontario, Canada

* amanda.formosa@mail.utoronto.ca (AF); claudia.santos@utoronto.ca (CCS)

**Data Availability Statement:** All relevant data are within the paper and its Supporting Information files.

## Abstract

The COVID-19 pandemic has created an urgency to study the host gene response that leads to variable clinical presentations of the disease, particularly the critical illness response. miRNAs have been implicated in the mechanism of host immune dysregulation and thus hold potential as biomarkers and/or therapeutic agents with clinical application. Hence, further analyses of their altered expression in COVID-19 is warranted. An important basis for this is identifying appropriate reference genes for high quality expression analysis studies. In the current report, NanoString technology was used to study the expression of 798 miRNAs in the peripheral blood of 24 critically ill patients, 12 had COVID-19 and 12 were COVID-19 negative. A list of potentially stable candidate reference genes was generated that included ten miRNAs. The top six were analyzed using reverse transcription quantitative polymerase chain reaction (RT-qPCR) in a total of 41 patients so as to apply standard computational algorithms for validating reference genes, namely geNorm, NormFinder, BestKeeper and RefFinder. There was general agreement among all four algorithms in the ranking of four stable miRNAs: miR-186-5p, miR-148b-3p, miR-194-5p and miR-448. A detailed analysis of their output rankings led to the conclusion that miR-186-5p and miR-148b-3p are appropriate reference genes for miRNA expression studies using PaxGene tubes in the peripheral blood of patients critically ill with COVID-19 disease.

**Funding:** A NanoString grant provided this study with reagents for experiments and technical support. The St. Michael's Hospital Foundation provided financial support to collect patient samples.

**Competing interests:** The authors have declared that no competing interests exist.

## Introduction

The World Health Organization declared the coronavirus disease 2019 (COVID-19) pandemic in March 2020 [1]. As of August 2022, COVID-19 has accounted for more than 577 million confirmed cases and more than six million confirmed deaths [2]. The course of illness of COVID-19 ranges from mild to severe and can be vastly unpredictable [3, 4], however certain groups are particularly vulnerable to severe illness such as the elderly and comorbid patient population [5]. Mortality rates in the literature range from 17–39% among critically ill patients [5]. While the advent of vaccination has had a remarkable effect in preventing serious illness [6, 7], vaccinated individuals can suffer breakthrough infections with high illness severity [4]. Given the variable clinical course of this disease, clinicians face challenges in assessing (i) a patient's vulnerability and (ii) deciding when and if to initiate appropriate therapeutic interventions.

Scientists have diverted a unique focus to understanding the dysregulated host immune response in COVID-19 responsible for variable disease phenotypes described above. Both the innate and adaptive immune responses contribute to an overactive inflammatory process, which may lead to end organ damage [8]. The virus enters respiratory epithelial cells through the ACE2 receptor and can evade intracellular immune responses and affect an altered type I interferon response [9, 10]. It is not yet known why some people have asymptomatic infection while others progress toward potentially lethal immune activation, however, genetic variants in immune components have been proposed [11, 12].

A complete picture on how COVID-19 leads to severe illness likely includes various mechanisms such as host immune response, inflammation and damage repair. Several studies have identified potential peripheral blood markers of COVID-19 related to those processes [13–16]. Other reports have turned to blood transcriptomics to gain large scale data on potential mechanisms attributable to COVID-19 outcomes [17–21]. Blood transcriptomics has emerged in the last decade as a rich source of biomarker discovery with applications in various fields including sepsis [22], cancer [23] and cardiovascular disease [24].

The non-coding RNAome is a key aspect of transcriptomic work alongside analyses of messenger RNA. miRNAs are endogenous non-coding RNA molecules of approximately 21 nucleotides in length [25]. They affect gene expression by binding to an mRNA's 3'-untranslated region (UTR) to degrade or inhibit protein translation [25]. The circulating non-coding transcriptome, including miRNAs, has been shown to provide beneficial information as a biomarker of disease diagnosis and prognostication for adverse outcomes, with potential for therapeutic intervention that has reached clinical trials [26–30].

An important aspect of high-quality transcriptomic work involves technical verification of differentially expressed nucleic acids using reverse transcription quantitative polymerase chain reaction (RT-qPCR) as a gold standard. Accurate and reliable results rely heavily on the selection of appropriate reference genes. Several computational programs have been designed to aid in the selection of stable housekeeping genes across a range of test and control samples for a given experimental context [31–34]. The importance of such programs as a starting point for achieving reliable results in differential expression analyses has been well established [35, 36].

In the current analysis, reference genes have been systematically selected for miRNA studies involving the peripheral blood of COVID-19 patients collected in PAXgene tubes. PAXgene tubes allow for ease of sample collection and processing with prolonged maintenance of nucleic acid stability, and as such their use in transcriptomic work is widespread [20]. The objective of this study was to 1) identify potential housekeeping genes across a range of clinical samples from patients with critical illness and 2) verify a subset of housekeeping genes for direct use in miRNA expression analyses in the particular clinical context presented here–critically unwell patients with and without COVID-19.

## Materials and methods

### Ethics and blood collection

Approval for the collection of peripheral blood from critically ill patients was obtained from the St. Michael's Hospital Research Ethics Board (REB number 20–078). Written consent from substitute decision makers was obtained for enrollment in our study as per institution protocol. Blood was collected within 48 hours of ICU admission into PAXgene Blood RNA Tubes (PreAnalytiX, Hombrechtikon, Switzerland, catalog number 762165) as per manufacturer instructions. Briefly, approximately 2.5mL of blood was drawn directly into the PAXgene tube and stored at -80˚C until analysis.

### Nucleic acid extraction

Before analysis, samples were thawed and kept at room temperature with PAXgene reagent for approximately 1.5hrs to increase yield. Total RNA, including miRNAs, was extracted using the PAXgene blood miRNA kit (PreAnalytiX catalogue number 763134) according to manufacturer instructions. RNA quantification was carried out with the Qubit 4 Fluorometer (Invitrogen, catalogue number Q33226) ThermoFisher Scientific, MA, USA, using the RNA HS assay kit (same company; product number Q32852). A subset of samples were analyzed using the Agilent 2100 Bioanalyzer (Agilent Technologies; catalogue number G2939BA), CA, USA, with the RNA 6000 NanoKit (same company, PN #5067–1511) to ensure a minimum RIN of 8.

### NanoString quantification

100ng of total RNA was used as an input for the NanoString nCounter human v3 miRNA expression panel (CSO-MIR3-24) (NanoString, Washington, United States), which profiles 798 human miRNAs. Samples (24 in total) were prepared as recommended under the nCounter miRNA Expression Assay with the following modification: an attenuation probe master mix was created using miR-451a (40nM) and miR-16-5p (4nM) attenuating oligonucleotides in order to remove 99% and 90% of these miRNAs, respectively. The oligo sequences were as follows: hsa-miR-451a – `GCCCATAGTTATTAATCTCTGTTTCAGCAATGAAAACCACCGCA AAGAAG`; hsa-miR-16-5p – `ATGCTAACGCTTTAGAGTATTTTGATGCGCGTTTAAAAGAGA TTTTAGAC`. Two microlitres of the Attenuation Probe Master Mix was mixed in with each sample after adding the Reporter/Hybridization mix.

### Data analysis, selection and validation of reference genes

Normalized counts for each gene per sample were generated with ROSALIND® NanoString miRNA Expression Methods (https://rosalind.onramp.bio/), whereby normalization is carried out using criteria as stated in the NanoString nSolver 4.0 User Manual.

Two methods were used to generate a list of candidate reference genes (Method 1 and Method 2). In Method 1, the standard deviation (STDEV) of normalized counts per gene across all 24 samples (analyzed in NanoString experiments) was computed. Given this study searches for appropriate reference genes and good expression in all samples is needed for a reference gene to be successful, only those genes expressed in all samples, with normalized counts greater than or equal to 8 were included. Then, a ranking of genes on the basis of their STDEV was done and the top three genes with the smallest STDEV were carried forward for validation assays.

For Method 2, candidate in silico miRNA controls were identified using NanoString.RCC files generated from the same 24 samples, which were read into R [37] using the NanoStringQCPro [38] package, and each sample was evaluated for quality control metrics related to imaging, binding density, limit of detection, and positive linearity of controls using the

NanoNormIter package [39]. All samples passed the quality control metric based on previously established guidelines [39]. As the housekeeping genes B2M (p<0.05), GAPDH (p<0.05), and ACTB (p<0.1) were found to be differentially expressed between COVID-19 + and COVID-19 –groups using a negative binomial generalized linear model, we sought to find in silico controls that were not differentially expressed. Upper quartile normalization was performed on the endogenous probes, followed by estimating the common and tagwise dispersions before running a negative binomial generalized log-linear model using functions from the edgeR package [40] to test for differential expression between COVID-19 + and COVID-19— patients. In silico controls were selected from 100 least differentially expressed probes, further filtered for a mean expression level of log2 > 6.5 (~90) across samples, and a coefficient of variation (CV) < = 0.50. Again, the top three selected probes across all samples, were carried forward in RT-qPCR assays.

## Reverse transcription quantitative PCR (RT-qPCR)

10ng of total RNA per sample were used for the reverse transcription reaction using the TaqMan Advanced miRNA cDNA Synthesis Kit (Applied Biosystems, Massachusettes, United States, A28007) as per manufacturer instructions. Real time PCR was carried out using TaqMan Advanced miRNA Assays (Applied Biosystems, A25576) and the TaqMan Fast Advanced Master Mix (Applied Biosystems, 4444963) on duplicate technical replicates for a total of 41 patients (please note these included the 24 nanostring patients) with and without COVID-19 as per manufacturer instructions, however, the final qPCR reaction was carried out in a total of 10uL of reaction volume (volumes for each reagent were scaled down proportionally). The sample maximization method was used, whereby all the patient samples and technical replicates were run in the same plate for a single gene to avoid inter-plate variability [41]. The assays used were (top 3 from Method 1 and top 3 from Method 2): hsa-miR-148b-3p, hsa-miR-2116-5p; hsa-miR-216b-5p; hsa-miR-448; hsa-miR-194-5p; hsa-miR-186-5p; Real time qPCR (RT-qPCR) was carried out using QuantStudio™ 7 Flex Real-Time PCR System, 384-well, desktop (Applied Biosystems, catalog number 4485701).

The averaged Ct values for technical replicates (Ct duplicate data less than or equal to 0.5 cycles only was included) per miRNA assay for each of 41 patient samples were input into Reffinder (accessed at http://blooge.cn/RefFinder/?type=reference) [42]. The Pearson correlation for technical replicates for the four candidate reference genes used for downstream analysis were as follows: hsa-miR-148b-3p (0.98), hsa-miR-448 (0.98), hsa-miR-194-5p (0.76) and hsa-miR-186-5p (0.95).

The averaged expression levels of the 4 candidate reference miRNAs were tested for associations with the following study population characteristics in the 41 patient cohort: sex, age, the presence of comorbidities (hypertension, type II diabetes, dyslipidemia) and treatments (antibiotics, steroid use, respiratory support). For binary metadata (2 groups), miRNA expression was compared using a Welch's 2-sided t-test (S1 Table). Categorical metadata with 3 or more groups was modelled against miRNA expression using ANOVA. Continuous parameters were modelled using linear regression (S2 Table). No significant associations (p < 0.05) were found between any reference miRNA and the study population characteristics tested.

## Software programs used for expression stability analysis of reference genes

The six candidate reference genes hsa-miR-148b-3p, hsa-miR-2116-5p, hsa-miR-216b-5p, hsa-miR-448, hsa-miR-194-5p and hsa-miR-186-5p were analyzed by RT-qPCR as described above in 41 patients with and without COVID-19 for their expression stability. Average Ct values for four candidate reference genes that had adequate expression in RT-qPCR (hsa-miR-

148b-3p, hsa-miR-448, hsa-miR-194-5p and hsa-miR-186-5p) were input into Genorm [33], Normfinder [32], BestKeeper [31] and RefFinder [42] (raw Ct data can be found in S3 Table).

GeNorm and NormFinder algorithms were used as implemented in the Bioconductor packages ReadqPCR v 1.42.0 and NormqPCR v 1.42.0 [43]. The quantification cycle (Ct) values (also known as the threshold-cycle (Ct) values) of technical replicates are imported into R (v 4.2.0) using the read.qPCR function to generate an r-object of class "qPCRBatch" to contain the miRNA gene expression and associated clinical metadata. To identify potential housekeeping genes, the selectHKs function from NormqPCR is subsequently used to find the optimum reference genes, with minimum number of housekeeping genes set to 2 and log = TRUE. With the geNorm method, stability values M is calculated sequentially to narrow down to a final set of two genes. For the NormFinder algorithm, the gene expression stability value rho is calculated on the same qPCRBatch object.

BestKeeper (version 1) was used through the Excel-based tool obtainable from https://www.gene-quantification.de/bestkeeper.html#download (accessed August 10th 2022).

The algorithm is based on the concept that lower variation in Ct values (when cDNA amount is constant) reflects higher gene expression stability. Ct values are used as input data and the algorithm then calculates various values, many of which are markers of variation in data and include: geometric (Geo Mean), arithmetic mean (Ar Mean), minimum and maximum Ct, Ct standard deviation (Std Dev) and coefficient of variation (CV). Ct values with Std Dev higher than 1 are considered inconsistent. The authors suggest using more than one reference gene in order to get more reliable results. For all possible pairs of reference genes, the algorithm performs pair-wise correlation analyses and for every calculated correlation, the Pearson correlation coefficient (r) and the probability p value are calculated. The highly correlated House Keeping Genes (HKGs) are combined into an index, and then a comparison is made between each candidate HKG and the index yielding a Pearson correlation coefficient (r), coefficient of determination (r2) and p-value.

RefFinder was used through http://blooge.cn/RefFinder/?type=reference (accessed on August 10th 2022). It provides a user-friendly web-based platform to analyze large datasets of reference genes and integrates the major computational programs (GeNorm, NormFinder, BestKeeper and the comparative delta Ct method) in its analysis. The output from each software program is then used to assign an appropriate weight to each gene, calculate the geomean of weights and use this to output an overall final ranking.

## Results

### Candidate reference gene selection

NanoString technology was used to profile 798 human miRNAs in the peripheral blood of 24 patients, 12 of which were COVID-19 positive and 12 which were negative.

The miRNA expression data was then processed in two distinct manners–Method 1 and Method 2 –so as to obtain an array of candidate reference genes to test in RT-qPCR and major computational programs that analyze reference genes. RT-qPCR/computational analysis was done in a total of 41 patients (of note, this included 16 of the NanoString patients already mentioned). Table 1 lists the top five miRNAs deemed as most stable by each method.

### Validation of candidate reference gene expression stability in the peripheral blood of patients with COVID-19

The top three stable miRNAs resulting from each of Method 1 and Method 2 were used in an RT-qPCR experiment to generate data that can be subject to appropriate reference gene

**Table 1. Five top ranking candidate miRNA reference genes generated by two separate methods using NanoString data.**

| Method 1 | Method 2 |
|---|---|
| Human miRNA | Human miRNA |
| miR-148b-3p | miR-2116-5p |
| miR-194-5p | miR-216b-5p |
| miR-186-5p | miR-448 |
| miR-331-3p | miR-944 |
| miR-30e-5p | miR-146a-5p |

validation by four gold standard computational algorithms. This was done in a total of 41 patients. Table 2 displays the patient characteristics.

The top six miRNAs, as shown in Table 2, were as follows: miR-148b-3p, miR-194-5p, miR-186-5p, miR-2116-5p, miR-216b-5p and miR-448. RT-qPCR analysis yielded poor results for miR-2116-5p and miR-216b-5p –a large number of samples had poor expression with "undetermined" values. Given that reference gene quality is incumbent on high expression in the sample population at hand, these two candidate reference genes were eliminated from further analysis. Samples were included in the analysis only if technical replicate Ct values were equal to or less than 0.5 cycles. For the candidate miRNAs carried forward for analysis by reference gene selection algorithms, the minimum and maximum values of averaged Ct values across all patients were as follows: miR-194-5p (20.02 to 24.76); miR-186-5p (20.96 to 25.01); miR-148b-3p (20.13 to 25.18); miR-448 (26.44 to 34.04).

Table 3 shows the reference gene rankings for GeNorm, NormFinder and BestKeeper, along with associated stability values. The accepted gene stability thresholds for the three algorithms are M<1.5, SV<1.0 and SD<1.0, respectively [44]. The GeNorm and NormFinder analysis suggests that all four reference genes selected would be considered stable enough, whereas the BestKeeper analysis shows that miR-448 would not meet the rejection cutoff (St Dev about 1), however, miR-186-5p, miR-194-5p and miR-148b-3p would be considered adequate. Importantly, the three algorithms mainly agreed on the ranking of gene stability–all three identified miR-186-5p as the top stable reference gene. GeNorm and BestKeeper both resulted in miR-148b-3p as the second-best ranking gene. All three algorithms placed miR-448 as the least stable reference gene.

The use of a single reference gene in experiments is discouraged given that is can introduce bias, and as such multiple reference genes are suggested as standard practice [33]. The pairwise variation (V) analysis calculated with geNorm helps to identify the ideal numbers of reference

**Table 2. Patient characteristics.**

| Characteristic | Nanostring cohort (n = 24) | RT-qPCR cohort (n = 41, including 16 patients from nanostring cohort) |
|---|---|---|
| Age, median years (IQR) | 60 (24.8) | 61 (21) |
| Sex, n (%) | | |
| Female | 9 (37.5) | 8 (19.5) |
| Male | 15 (62.5) | 33 (80.4) |
| COVID-19 status, n (%) | | |
| Positive | 12 (50) | 20 (51.7) |
| Negative | 12 (50) | 21 (48.3) |
| Initial SOFA score (within 48 hours of admission), mean (STDEV) | 7.7 (4.1) | 8.3 (4.0) |

**Table 3. Reference gene ranking by GeNorm, NormFinder and BestKeeper and associated stability values (from most to least stable).**

| GeNorm | GeNorm M value | NormFinder | NormFinder Stability value rho | BestKeeper | BestKeeper Std Dev |
|---|---|---|---|---|---|
| miR-186-5p/miR-148b-3p | 0.536 | miR-186-5p | 0.129 | miR-186-5p | 0.86 |
| miR-194-5p | 0.670 | miR-194-5p | 0.151 | miR-148b-3p | 0.9 |
| miR-448 | 1.11 | miR-148b-3p | 0.286 | miR-194-5p | 0.9 |
| | | miR-448 | 0.447 | miR-448 | 1.15 |

genes to be used (a lower V number means including the additional reference gene in question is beneficial). The ideal cutoff value of V <0.15 determines when to stop adding additional reference genes. However, this number is considered to be a suggested threshold criteria rather than a universal cutoff and the experimental context should be taken into account, referring mainly to the biological system at hand and the level of variation expected (for example, less variable cell line extraction data versus animal tissues or human biological samples) [45]. The samples under analysis in this study are highly variable and thus we considered the V2/V3 at 0.229 to be sufficient, as V3/V4 was higher at 0.377. Several studies have also indicated the use of slightly higher V values [45–48]. Chasing more reference genes to target a perfect V value when these results are near to the cutoff would be highly impractical, since the experiments described here require a large throughput of data in many patient samples, oftentimes including multiple target genes. Having greater than four reference genes would be expensive and labor intensive for likely no or little benefit in accuracy of data processing.

Table 4 shows the output data for BestKeeper, which assigns higher stability to lower Ct variation. The most stable reference genes are selected on the basis of the lowest Ct standard deviation values [31, 49]. As already alluded to, all reference genes except miR-448 met the cutoff criteria of 1 for stability. Furthermore, the software produces pair-wise correlation results to estimate the relationship between reference gene pairs (data not shown). P-values for all reference genes are less than 0.05 are considered to be correlated with the BestKeeper index and hence are good reference genes.

Lastly, raw Ct data were analyzed by RefFinder software. Fig 1 displays the output graphics from RefFinder, illustrating results in agreement with those described above for the previous three algorithms. The comprehensive ranking also yields highly agreeable results, with miR-186-5p and miR-148b-3p as the top two reference genes.

## Discussion

Identifying stable reference genes is of the utmost importance in correctly interpreting gene expression in any experimental design. The choice of a wrong reference gene can greatly affect the target gene results, and the use of multiple reference genes is considered to be important

**Table 4. Analysis of reference genes by BestKeeper algorithm.**

| | miR-194-5p | miR-186-5p | miR-148b-3p | miR-448 |
|---|---|---|---|---|
| n | 41 | 41 | 41 | 41 |
| geo Mean [CP] | 22.29 | 22.77 | 22.54 | 31.01 |
| ar Mean [CP] | 22.32 | 22.79 | 22.56 | 31.05 |
| min [CP] | 20.02 | 20.96 | 20.13 | 26.45 |
| max [CP} | 24.76 | 25.01 | 25.18 | 34.04 |
| std dev [+/- CP] | 0.90 | 0.86 | 0.90 | 1.15 |
| CV [% CP] | 4.05 | 3.77 | 3.97 | 3.70 |
| p-value | 0.001 | 0.001 | 0.001 | 0.001 |

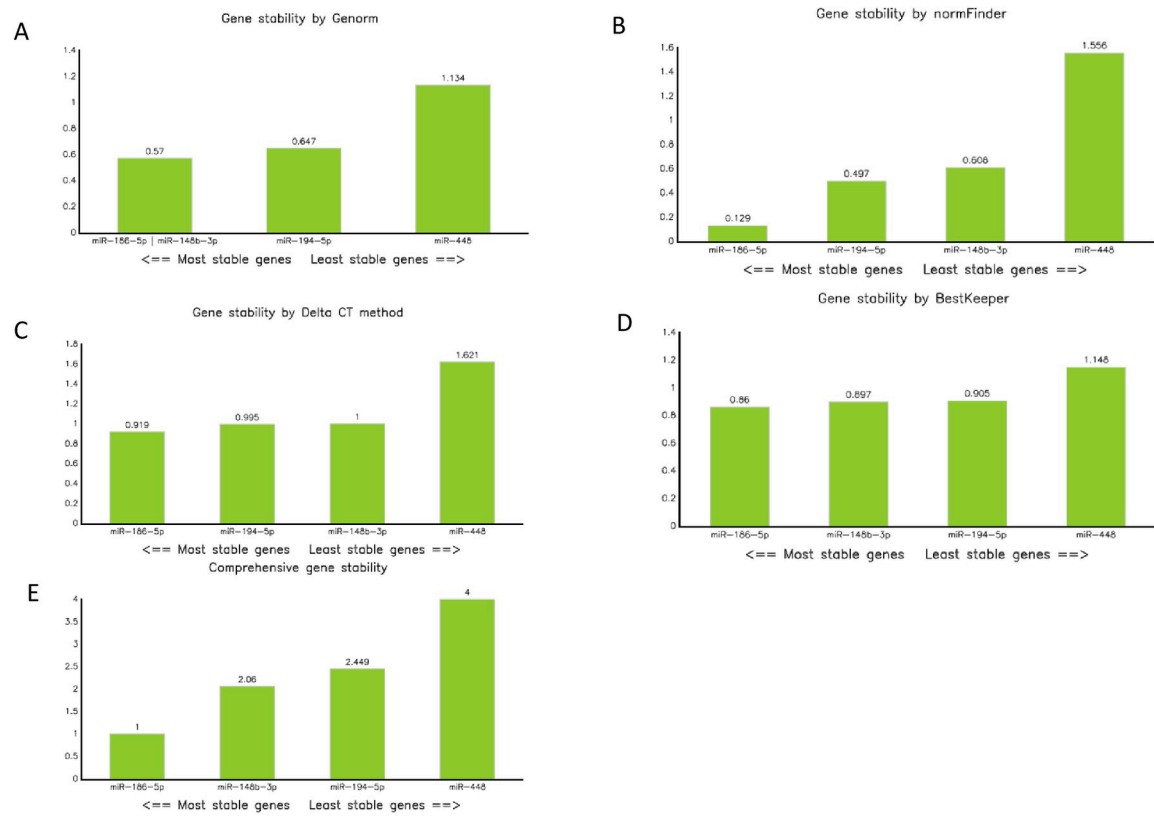

**Fig 1. RefFinder output for candidate reference genes across 41 patients.** (A) geNorm ranking by stability value (B) normFinder ranking by stability value (C) delta CT method ranking by average STDEV (D) BestKeeper std dev ranking and (E) comprehensive ranking (geomean of ranking values).

for accurate analysis [33, 50]. The current report sought to identify reliable reference genes for use in RT-qPCR miRNA expression studies in the peripheral blood of critically ill patients with and without COVID-19. The results are particularly applicable for blood collected in Pax-Gene tubes, which is becoming an increasing common way to sample human specimens due to its ease of use and reliability in preserving nucleic acid species [21, 51, 52]. The methodology employed here, with the use of geNorm, NormFinder, BestKeeper and RefFinder, represents a comprehensive review of standard approaches used numerous times in the literature to reliably identify appropriate reference genes [44, 45, 49].

The search for reference genes for a new experimental design usually involves a detailed literature search to obtain a list of candidate genes used in similar studies that are then formally tested in the samples at hand. Given that our cohort is unique in several ways (COVID negative patients as controls and whole blood processed in PaxGene tubes), we did not feel our original literature search would produce an adequate list of candidate reference genes that would have a high chance of success in standard reference gene computational analyses. Thus, the first part of our analysis involved a large-scale study of the expression of 798 miRNAs in a smaller cohort of patients using NanoString technology. This was carried out to obtain a list of six potential reference gene miRNAs for appropriate testing in RT-qPCR/computational program analysis: miR-186-5p, miR-148b-3p, miR-194-5p, miR-2116-5p, miR-216b-5p and miR-448. Two of the miRNAs, miR-2116-5p and miR-216b-5p, had low expression in RT-qPCR analysis and thus the associated expression data was not carried forward.

Four miRNAs have been deemed stable enough as reference genes by most of the computational programs employed here: miR-186-5p, miR-148b-3p, miR-194-5p and miR-448. To the best of our knowledge, none of these candidates have been underscored as biomarkers or significantly deregulated in COVID-19 critical illness. One candidate (miR-186-5p) has been shown in one study to be decreased in the plasma of COVID-19 patients, however, the study population was small and very different to what we present here: COVID-19 patients (n = 12) were mainly mildly ill and results were compared to healthy controls [53]. Our results suggest that the top two miRNAs, miR-186-5p and miR-148b-3p, used in conjunction (as suggested by GeNorm technology) offer the best means to normalize target gene expression data. Importantly, all software programs were in agreement the vast majority of the time on the ranking of stable reference genes, which suggests that our experimental design and data analysis was robust. There was a minor difference in ranking with NormFinder, in that the top two reference genes listed were miR-186-5p and miR-194-5p. Minor differences in ranking across algorithms are not unexpected, since their approaches have inherent differences, as described above [31–33, 42]. Candidate reference miRNAs were not significantly associated ($p < 0.05$) with the study population characteristics of sex, age, comorbidities (type II diabetes, hypertension, dyslipidemia) or treatments (antibiotics, steroid use, respiratory support) (S1, S2 Tables).

This study has certain limitations. The use of paxgene tubes provide a high level of convenience for obtaining peripheral blood at the bedside. The caveat is that the whole sample is immersed in a proprietary reagent which lyses cells and stabilizes nucleic acids and as such the source of miRNA signal is not known. Our choice of a COVID-negative critically ill control group was deliberate in that it offers the most clinically relevant comparison. However, many studies continue to use healthy controls as a comparison group and in that regard the utility of applying these miRNAs candidates in that particular experimental scenario is unknown. We would like to re-iterate that while the reference genes identified here are appropriately validated for our sample population, they should be considered *candidate* reference genes that need validation for any new experimental context, which may indeed include healthy controls or different sample types such as plasma or serum.

The reference genes identified here can likely be considered appropriate candidates for miRNA studies in non-COVID-19 critical illness scenarios, since the non-COVID-19 group represent heterogenous diseases lending to Intensive Care Unit admission, such as acute respiratory distress syndrome and sepsis. The initial phase of creating an appropriate list of potential miRNAs is expensive and time consuming, and the current report may offer authors a means to bypass that screening step by offering ten reasonable reference gene choices as shown in Table 1. Again, we strongly suggest verifying such candidates in the particular experimental analysis under consideration.

## Conclusion

The COVID-19 pandemic has created an urgency to study the host gene response that leads to variable clinical presentations of the disease, particularly the critical illness response. miRNAs have been implicated in the mechanism of host immune dysregulation thus necessitating further analyses of their altered expression in COVID-19, and an important basis for this is identifying appropriate reference genes for high quality expression analysis studies. The current report aimed to identify reference genes for use in miRNA expression analysis in the peripheral blood of critically ill patients with and without COVID-19 disease. A list of ten candidate reference genes was generated using NanoString technology and six were investigated in validation studies by RT-qPCR. We have validated four stably expressed miRNAs and conclude that miR-186-5p and miR-148b-3p used in combination are appropriate reference genes for

miRNA expression studies using PaxGene tubes in the peripheral blood of patients critically ill with COVID-19 disease.

## Supporting information

**S1 Table. MiRNA expression analysis across comorbidities for binary metadata using Welch's two sided T test.**
(XLSX)

**S2 Table. MiRNA expression analysis across comorbidities for continuous metadata using linear regression.**
(XLSX)

**S3 Table. Data for qPCR technical replicates used as input data for algorithms.**
(XLSX)

**S4 Table. Normalized counts for the 5 candidate miRNAs selected from Nanostring.**
(XLSX)

## Acknowledgments

We thank our research coordinators Gyan Sandhu and Dr. Valeria Di Giovanni for their contributions to human sample collection and organizing of our biobank, respectively. We also thank patients that donated samples for this project.

## Author Contributions

**Conceptualization:** Amanda Formosa, Amy Lee, Uriel Trahtemberg, Andrew Baker, Claudia C. dos Santos.

**Data curation:** Amanda Formosa, Erica Acton, Amy Lee, Paul Turgeon, Shehla Izhar, Pamela Plant, Uriel Trahtemberg, Claudia C. dos Santos.

**Formal analysis:** Amanda Formosa, Erica Acton, Amy Lee, Pamela Plant, Sabri Soussi, Claudia C. dos Santos.

**Funding acquisition:** Amy Lee, Uriel Trahtemberg, Andrew Baker, Claudia C. dos Santos.

**Investigation:** Shehla Izhar, Jim N. Tsoporis, Andrew Baker, Claudia C. dos Santos.

**Methodology:** Amanda Formosa, Erica Acton, Paul Turgeon, Shehla Izhar, Pamela Plant, Jim N. Tsoporis.

**Project administration:** Uriel Trahtemberg, Andrew Baker, Claudia C. dos Santos.

**Resources:** Andrew Baker, Claudia C. dos Santos.

**Supervision:** Amy Lee, Claudia C. dos Santos.

**Writing – original draft:** Amanda Formosa.

**Writing – review & editing:** Amanda Formosa, Jim N. Tsoporis, Sabri Soussi, Claudia C. dos Santos.

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
