## [Decision Letter · Decision Letter 0]

16 Mar 2023

PONE-D-23-03677Validation of reference gene stability for miRNA quantification by reverse transcription quantitative PCR in the peripheral blood of patients with COVID-19 critical illnessPLOS ONE

Dear Dr. Formosa,

Thank you for submitting your manuscript to PLOS ONE. After careful consideration, we feel that it has merit but does not fully meet PLOS ONE’s publication criteria as it currently stands. Therefore, we invite you to submit a revised version of the manuscript that addresses the points raised during the review process.

We look forward to receiving your revised manuscript.

Kind regards,

Jacopo Sabbatinelli, MD, PhD

Academic Editor

PLOS ONE

Journal Requirements:

Reviewers' comments:

Reviewer's Responses to Questions

**Comments to the Author**

1. Is the manuscript technically sound, and do the data support the conclusions?

Reviewer #1: Yes

2. Has the statistical analysis been performed appropriately and rigorously? 

Reviewer #1: No

3. Have the authors made all data underlying the findings in their manuscript fully available?

Reviewer #1: Yes

4. Is the manuscript presented in an intelligible fashion and written in standard English?

Reviewer #1: Yes

5. Review Comments to the Author

Reviewer #1: Formosa et al. explore potential candidate as endogenous controls for miRNA evaluation in the critically ill patients. The manuscript is of interest. Some comments:

Abstract. I would highlight the potential of miRNAs as biomarkers with potential clinical application. One of the consequences of your results is the translation to the clinical practice could be facilitated with the miRNA candidates.

Introduction. I would highlight the impact of severe COVID-19 in more vulnerable groups elderly patients, with high level of comorbidities, immunodeficiency, ... Please, include some data on the high rate of mortality among critically ill patients.

In addition, I would not only focus on the immune response. The adverse outcomes are caused by different mechanisms, immune response, inflammation, poor repair, ... Due to its nature, miRNAs would provide information on all these mechanisms. I would also introduce the use of miRNAs as biomarkers for adverse outcomes. The comparison COVID-19 vs healthy controls is not clinically relevant.

Please, provide additional data on the criteria used. For instance, what percentage of expression, related to the number of samples, did you establish? It should 100 % for both study phases. Did you establish a maximum Cq?

Authors should compare the levels of miRNAs according to the characteristics of the study population. Are there differences between sex, presence of comorbidities (hypertension, diabetes, …), treatments (corticoids, ventilatory support, …)? Do miRNAs correlate with age?

Discussion should be completed with different topics. Are these candidates biomarkers in COVID-19? Could these miRNAs be quantified in other type of samples such as serum or plasma?

Could you evaluate the impact of the hemolysis in your samples?

“RNA concentration of the samples was determined using ThermoFisher Qubit system with RNA 140 HS assay kit.” It is repeated.

Which criteria did you use to evaluate duplicates? When did you repeat the qPCR?

Due to the huge problem in reproducibility, please, provide detailed information on the RNA isolation and RT-qPCR protocols. At least as supplemental information.

Please, use the appropriate statistical test to compare both cohorts.

I would add a Limitations paragraph

Please, add additional information to the Figure Legend.

6. PLOS authors have the option to publish the peer review history of their article (what does this mean?). If published, this will include your full peer review and any attached files.

Reviewer #1: No

---

## [Author Response · Author response to Decision Letter 0]

12 May 2023

May 9th 2023

Dear PLOS ONE editorial board,

Please find below our detail point by point response to the editor and reviewer’s comment for our manuscript entitled, “Validation of reference gene stability for miRNA quantification by reverse transcription quantitative PCR in the peripheral blood of patients with COVID-19 critical illness”. The editor/reviewer’s comments are listed and our response is directly below each comment. Please note that line numbers refer to the revised manuscript with track changes.

Response: we have accessed both urls and edited our manuscript in detail to reflect the formatting guidelines

Response: we obtained written consent from substitute decision makers and this is now updated in the methods section (lines 185-186)

Response: we agree that data sharing provides the best way for readers to fully assess and reproduce our results. The data is now included in Supplementary Tables (S1 – S4 Table) and updated in the manuscript file (lines 724-730)

Response: we now have our data compiled in Supplementary Tables S1-S4. 

5. PLOS requires an ORCID iD for the corresponding author in Editorial Manager on papers submitted after December 6th, 2016. Please ensure that you have an ORCID iD and that it is validated in Editorial Manager. To do this, go to ‘Update my Information’ (in the upper left-hand corner of the main menu), and click on the Fetch/Validate link next to the ORCID field. This will take you to the ORCID site and allow you to create a new iD or authenticate a pre-existing iD in Editorial Manager. Please see the following video for instructions on linking an ORCID iD to your Editorial Manager account:

Response: the ORCID ID for the submitting and corresponding author (Dr. A. Formosa) has been validated. I (A. Formosa) have contacted the editorial office regarding adding a second corresponding author, Dr. dos Santos. I was advised a second corresponding author is permitted and this is now reflected in the manuscript. I did not find a way to insert Dr. dos Santos’ ORCID ID in the system but it is: https://orcid.org/0000-0002-6446-8791

Reviewer #1: Formosa et al. explore potential candidate as endogenous controls for miRNA evaluation in the critically ill patients. The manuscript is of interest. Some comments:

Abstract. I would highlight the potential of miRNAs as biomarkers with potential clinical application. One of the consequences of your results is the translation to the clinical practice could be facilitated with the miRNA candidates.

Response: Thank you for this comment and the abstract has been adjusted accordingly (lines 71-72)

Introduction. I would highlight the impact of severe COVID-19 in more vulnerable groups elderly patients, with high level of comorbidities, immunodeficiency, ... Please, include some data on the high rate of mortality among critically ill patients.

Response: This information has now been included with a relevant citation (lines 109-111)

In addition, I would not only focus on the immune response. The adverse outcomes are caused by different mechanisms, immune response, inflammation, poor repair, ... Due to its nature, miRNAs would provide information on all these mechanisms. I would also introduce the use of miRNAs as biomarkers for adverse outcomes. The comparison COVID-19 vs healthy controls is not clinically relevant.

Response: Thank you for this comment and we agree. Our introduction has been edited accordingly and we have deleted the comment on comparison to healthy controls, which we also believe to be clinically irrelevant (lines 128-130; 135-140)

Please, provide additional data on the criteria used. For instance, what percentage of expression, related to the number of samples, did you establish? It should 100 % for both study phases. Did you establish a maximum Cq?

Response: We agree that expression in all samples is of utmost importance given we are studying reference genes. We have alluded to that in the methods when discussing the screening of candidate genes for RT-qPCR. To highlight this importance further, we have edited lines 236-238. For candidate miRNAs included in reference gene selection algorithms, only those with 100% expression across all patient samples were included. We have now added more information on this, including minimum and maximum Cq values, in our results section (lines 401-403). 

Authors should compare the levels of miRNAs according to the characteristics of the study population. Are there differences between sex, presence of comorbidities (hypertension, diabetes, …), treatments (corticoids, ventilatory support, …)? Do miRNAs correlate with age?

Response: The Ct values of the 4 candidate reference miRNAs (each averaged over two technical replicates) were tested for associations with the following study population characteristics in the validation cohort: sex, age, the presence of comorbidities (hypertension, type II diabetes, dyslipidemia) and treatments (antibiotics, steroid use, respiratory support). For binary metadata (2 groups), miRNA expression was compared using a Welch's 2-sided t-test. Categorical metadata with 3 or more groups was modelled against miRNA expression using ANOVA. Continuous parameters were modelled using linear regression. No significant associations (p < 0.05) were found between any reference miRNA and the study population characteristics tested. The model outputs and p-values are now included in Supplemental Table S1. The description of these tests are now included in the Methods at lines 282-294, and referenced in the Discussion at lines 525-527.

Discussion should be completed with different topics. Are these candidates biomarkers in COVID-19? 

Response: we reviewed the literature and did not find any evidence of these miRNAs as biomarkers in COVID-19 critical illness. This is now outlined in the discussion lines 495-500.

Could these miRNAs be quantified in other type of samples such as serum or plasma?

Response: we have commented on the potential of these miRNAs to be tested as candidate reference genes in different sample types in our new limitations paragraph (lines 529-539)

Could you evaluate the impact of the hemolysis in your samples?

Response: our new limitations paragraph addresses this concern in not knowing the origin of the miRNA signals (inherent issue to using Paxgene tubes, lines 530-532)

“RNA concentration of the samples was determined using ThermoFisher Qubit system with RNA 140 HS assay kit.” It is repeated.

Response: thank you for pointing this out. The redundant phrase has been deleted (lines 205-211).

Which criteria did you use to evaluate duplicates? When did you repeat the qPCR?

Response: duplicates were average only if Ct data were within 0.5 cycles of each other. All technical replicates for all samples for a given gene (sample maximization method) were run on the same qPCR plate. This is now noted in the materials and methods sections lines 267-268 and 280-281. 

Due to the huge problem in reproducibility, please, provide detailed information on the RNA isolation and RT-qPCR protocols. At least as supplemental information.

Response: we have peeled through our materials and methods section and added more detailed information to ensure reproducibility (lines 184-281)

Please, use the appropriate statistical test to compare both cohorts.

Response: we have included further statistical testing as requested above including the analysis of the four candidate reference miRNAs across population study characteristics. Our input data for GeNorm, Normfinder, Bestkeeper and RefFinder was Ct values across all patients as required by the algorithms (S3 – S4 tables).

I would add a Limitations paragraph

Response: this has been added (lines 529-539)

Please, add additional information to the Figure Legend.

Response: further information regarding the rankings and Y axis values has been added (lines 461-464)

---

## [Decision Letter · Decision Letter 1]

25 May 2023

Validation of reference gene stability for miRNA quantification by reverse transcription quantitative PCR in the peripheral blood of patients with COVID-19 critical illness

PONE-D-23-03677R1

Dear Dr. Formosa,

We’re pleased to inform you that your manuscript has been judged scientifically suitable for publication and will be formally accepted for publication once it meets all outstanding technical requirements.

Kind regards,

Jacopo Sabbatinelli, MD, PhD

Academic Editor

PLOS ONE

Additional Editor Comments (optional):

Reviewers' comments:

Reviewer's Responses to Questions

**Comments to the Author**

1. If the authors have adequately addressed your comments raised in a previous round of review and you feel that this manuscript is now acceptable for publication, you may indicate that here to bypass the “Comments to the Author” section, enter your conflict of interest statement in the “Confidential to Editor” section, and submit your "Accept" recommendation.

Reviewer #1: (No Response)

2. Is the manuscript technically sound, and do the data support the conclusions?

Reviewer #1: Yes

3. Has the statistical analysis been performed appropriately and rigorously? 

Reviewer #1: Yes

4. Have the authors made all data underlying the findings in their manuscript fully available?

Reviewer #1: Yes

5. Is the manuscript presented in an intelligible fashion and written in standard English?

Reviewer #1: Yes

6. Review Comments to the Author

Reviewer #1: The authors have properly addressed all comments. The manuscript has been substantially improved and deserves publication.

7. PLOS authors have the option to publish the peer review history of their article (what does this mean?). If published, this will include your full peer review and any attached files.

Reviewer #1: No

---

## [Editor Report · Acceptance letter]

21 Aug 2023

PONE-D-23-03677R1 

Validation of reference gene stability for miRNA quantification by reverse transcription quantitative PCR in the peripheral blood of patients with COVID-19 critical illness 

Dear Dr. Formosa:

I'm pleased to inform you that your manuscript has been deemed suitable for publication in PLOS ONE. Congratulations! Your manuscript is now with our production department. 

Kind regards, 

on behalf of

Dr. Jacopo Sabbatinelli 

Academic Editor

PLOS ONE